# Intestinal Carriage of Two Distinct *stx_2f_*-Carrying *Escherichia coli* Strains by a Child with Uncomplicated Diarrhea

**DOI:** 10.3390/pathogens13111002

**Published:** 2024-11-15

**Authors:** Florence Crombé, Angela H. A. M. van Hoek, Heleen Nailis, Frédéric Auvray, Toon Janssen, Denis Piérard

**Affiliations:** 1Department Clinical Biology, Laboratory of Microbiology and Infection Control, Belgian National Reference Centre for STEC/VTEC, Vrije Universiteit Brussel (VUB), Universitair Ziekenhuis Brussel (UZ Brussel), 1090 Brussels, Belgium; denis.pierard@uzbrussel.be; 2Centre for Infectious Disease Control (CIb), National Institute for Public Health and the Environment (RIVM), 3721 BA Bilthoven, The Netherlands; angela.van.hoek@rivm.nl; 3Department of Laboratory Medicine, AZ Turnhout, 2300 Turnhout, AZ Herentals, 2200 Herentals, Heilig Hart Mol, 2400 Mol, Belgium; heleen.nailis@azturnhout.be; 4IRSD, Faculté des Sciences, Université de Toulouse, INSERM, INRAE, ENVT, UPS, 31000 Toulouse, France; frederic.auvray@envt.fr; 5Brussels Interuniversity Genomics High Throughput Core (BRIGHTcore) Platform, Vrije Universiteit Brussel (VUB), Universitair Ziekenhuis Brussel (UZ Brussel), 1090 Brussels, Belgium; toon.janssen@uzbrussel.be

**Keywords:** Shiga toxin-producing *Escherichia coli* (STEC), Stx2f, *Escherichia coli* O157:H16, whole-genome sequencing

## Abstract

Two distinct *stx_2f_*-carrying *Escherichia coli* (*E. coli*) strains, isolated from a child with uncomplicated diarrhea fifteen weeks apart, were characterized by combining short- and long-read sequencing to compare their genetic relatedness. One strain was characterized as Shiga toxin-producing *E. coli* (STEC)/typical enteropathogenic *E. coli* (tEPEC) O63:H6 with a repertoire of virulence genes including *stx_2f_*, *eae* (α2-subtype), *cdt*, and *bfpA*. The other STEC with serotype O157:H16, reported for the first time as *stx_2f_*-carrying *Escherichia coli* in this study, possessed, in addition, *eae* (ε-subtype) and *cdt*, amongst other virulence-related genes. BLAST comparison showed that the *stx_2f_*-harboring prophage sequences of both strains were highly homologous (99.6% identity and 96.1% coverage). These results were corroborated by core Stx2f phage Multilocus Sequence Typing (cpMLST) as the *stx_2f_*-harboring prophages of both isolates clustered together when compared to those of 167 other human *stx_2f_*-carrying *Escherichia coli*. Overall, the *stx_2f_*-harboring prophages of the two distinct *E. coli* strains isolated from the present case were highly similar, suggesting that the *stx_2f_*-harboring phage might have been transferred from the STEC/tEPEC O63:H6 strain to the atypical EPEC (aEPEC) O157:H16 strain in the gut of the child.

## 1. Introduction

Shiga toxin-producing *Escherichia coli* (STEC) are causing a wide spectrum of gastrointestinal symptoms in humans, ranging from uncomplicated forms of intestinal illnesses to bloody diarrhea and life-threatening hemolytic uremic syndrome (HUS) [1]. The production of Shiga toxin (Stx1 and/or Stx2 subtypes, i.e., 1a, 1c–1e, 2a–2o), Stx2 in particular, is a major virulence factor associated with the development of severe symptoms [2]. However, the Stx2f subtype has rarely been isolated from patients with HUS and is generally linked to mild gastrointestinal symptoms [3,4].

In Europe, a substantial proportion of STEC strains involved in human infections are carrying *stx_2f_* [3,5,6]. The number of cases might even be underreported, as some molecular gastrointestinal assays for the laboratory diagnosis of STEC infection do not detect the *stx_2f_* gene, encoding the most divergent Stx2 subtype (by nucleotide and amino acid sequence) [4,7]. In Belgium, *stx_2f_*-carrying *E. coli* consisted of 12.1% (76/626) of all culture-positive cases of STEC infection in the period 2021–2023, with the highest number recorded in 2023 (58/331) [8]. During this period, the dominant *stx_2f_*-positive *E. coli* serotype was O63:H6 (67.1%; 51/76). The same was observed in 2021 in the European Union/European Economic Area (EU/EEA) where 47.0% of the *stx_2f_*-carrying *E. coli* isolates were serotyped as O63:H6 (data from 23 EU/EEA countries) [9]. Remarkably, this serotype has mainly been reported in humans [5,10] while other serotypes of *stx_2f_*-carrying *E. coli* have been described in pigeons [10,11] and other bird species [12].

The genes encoding the Shiga toxins (*stx*) are harbored by bacteriophages, which can be integrated as prophages within the chromosome of a susceptible *E. coli* strain or another member of *Enterobacteriaceae*, upgrading, for example, a non-pathogenic *E. coli* to a pathogenic STEC [13]. Several chromosomal insertion sites for Stx phages have been reported, including *ssrA*, which encodes a transfer-messenger RNA (tmRNA) [14]. This gene has been described as a typical insertion site for Stx2f phages in *E. coli* and other *Enterobacteriaceae*, including *E. albertii* [15,16].

In addition to *stx*, a subset of STEC strains possess the locus of enterocyte effacement (LEE) pathogenicity island (PAI), which contains genes that mediate colonization of the human intestine [17]. These genes encode type III secretion system proteins (T3SS), the intimin protein (Eae) and its translocated receptor (Tir), as well as chaperones, regulators, and secreted effector proteins (Esp). The LEE PAI is also found in enteropathogenic *E. coli* (EPEC) [18]. Yet, EPEC can be divided into typical EPEC (tEPEC) and atypical EPEC (aEPEC) based on the presence or absence, respectively, of the *E. coli* adherence factor plasmid carrying the bundle-forming pilus (BFP) operon; tEPEC are commonly *stx*-negative, LEE-positive, and BFP-positive, while aEPEC are also *stx*-negative and LEE-positive but BFP-negative. Consequently, the latter are thought to be of concern as they can acquire the Stx phages [19].

LEE-positive *E. coli* strains of O157:H16 serotype are usually aEPEC [20,21,22,23,24,25,26,27,28,29] (Appendix A). Strains of serotype O157:H16 have been reported to cause sporadic cases and outbreaks of diarrheal diseases [23,30]. This serotype has also been isolated from beef, cattle, dogs, and water [22,31]. Until now, only one *stx*-positive O157:H16 strain isolated from cattle has been reported [32].

In the present study, we aimed to characterize two distinct *stx_2f_*-carrying *E. coli* strains isolated from a child with uncomplicated diarrhea in order to compare their relatedness at the whole-genome level.

## 2. Materials and Methods

### 2.1. Case Description

A 17-month-old child presented at the pediatrics department of the AZ Turnhout on 18 October 2023 with symptoms of intermittent diarrhea for 10 days and rhinitis with fluctuating fever. The patient was treated symptomatically for viral infection. A fecal sample was collected on 26 October 2023.

On 6 November 2023, the child was diagnosed with a viral cough. A stool sample was taken the same day due to intermittent stool consistency.

Fifteen weeks later, the child consulted the pediatrician with complaints of intermittent vomiting, diarrhea for 4 days, and stomach cramps. The child was treated with probiotic Enterol^®^ (*Saccharomyces boulardii*) and oral rehydration solution. The child returned to the pediatrician five days later with complaints of watery diarrhea. The child was treated symptomatically with antidiarrheal Tiorfix^®^ (10 mg racecadotril/sachet) and recovered promptly. A fecal sample was collected on 12 February 2024.

### 2.2. Bacterial Isolates

At the AZ Turnhout, fecal specimens from patients presented to the pediatrician with symptoms of gastrointestinal infection are routinely tested for a range of gastrointestinal pathogens with the Allplex^TM^ Gastrointestinal Panel Assays (Seegene, Seoul, Republic of Korea), i.e., Allplex^TM^ GI-EB Screening Assay, Allplex^TM^ GI-Virus Assay, and Allplex^TM^ GI-parasite Assay. Fecal specimens tested positive for *stx* by the Allplex^TM^ GI-EB Assay are referred to the NRC STEC for culture.

At the NRC STEC, all fecal specimens are cultured on to sorbitol MacConkey agar bi-plates with and without cefixime (0.05 mg/L) and tellurite (2.5 mg/L) (SMAC/CT-SMAC), and individual colonies are tested using in-house polymerase chain reaction (PCR) assays targeting *stx_1_*, *stx_2_* (including *stx_2f_*), and *eae* [3]. All STEC isolates are serotyped by slide agglutination. To that end, the *E. coli* OK O Pool 1 antisera (anti-O26, O103, O111, O145, and O157; Statens Serum Institut, Copenhagen, Denmark) is used. In the case of a negative reaction, the serogroup of the isolate is defined as non-O157. Antimicrobial susceptibility testing is performed on all STEC isolates by disk diffusion according to EUCAST [33]. The panel of antimicrobials includes ampicillin, amoxicillin-clavulanic acid, piperacillin-tazobactam, cefadroxil, cefuroxime, ceftriaxone, ceftazidime, cefepime, aztreonam, temocillin, meropenem, ciprofloxacin, gentamicin, amikacin, and trimethoprim-sulfamethoxazole. All STEC isolates are analyzed with whole-genome sequencing.

### 2.3. DNA Extraction and Whole-Genome Sequencing

Genomic DNA was extracted from pure cultures of *E. coli* isolates grown overnight on Sorbitol MacConkey Agar (Neogen, Lansing, MI, USA) by using the Maxwell RSC Cell DNA purification kit (Promega Corporation, Madison, WI, USA) according to the manufacturer’s instructions.

#### 2.3.1. Illumina Sequencing

Fragmentation of 500 ng of genomic DNA was carried out using the NEBNext^®^ Ultra^TM^ II FS module. Sequencing libraries, with an insert size of 550 bp on average, were prepared using the KAPA Hyper Plus kit (Kapa Biosystems, Wilmington, NC, USA). To avoid PCR bias, the PCR amplification step was omitted, and every sample was assigned an in-house Truseq style adapter with a unique dual indexed 8-bp barcode. After equimolar pooling, libraries were sequenced on a Novaseq 6000 instrument (Illumina, San Diego, CA, USA) using the NovaSeq 6000 SP Reagent kit (300 cycles) generating 2 × 150 bp reads. For this, the library was denaturated and diluted according to the manufacturer’s instructions. A 1% PhiX control library was included in each sequencing run.

#### 2.3.2. Oxford NanoPore Sequencing

In parallel, the sequencing libraries were prepared using the Rapid barcoding kit v14 (SQK-RBK114.96, Oxford Nanopore Technology, Oxford, UK) and sequenced for 12 h on a PromethION R1041 flowcell (FLO-PRO114M, Oxford Nanopore Technology, Oxford, UK). Sequencing was performed using MinKNOW 23.07.5, and basecalling was performed using Guppy 7.0.9.

### 2.4. In Silico Multilocus Sequence Typing Analysis and Identification of Genes Linked to Serotype, Virulence, and Antibiotic Resistance

The raw reads were imported and *de novo* hybrid-assembled using Unicycler 0.5.0 (normal mode) in the Galaxy Europe platform [34]. Subsequent prediction of sequence types (STs), serotypes, and acquired antibiotic resistance genes was performed using tools available from the Center for Genomic Epidemiology platform (https://www.genomicepidemiology.org/services/ (accessed on 14 June 2024)) (MLST 2.0, SerotypeFinder 2.0 and ResFinder 4.5.0). A % identity threshold of 85% and a minimum length for coverage of 60% was used. The phylogroups were predicted using the Clermont Typing online tool [35]. The virulence gene profiles were predicted starting from the genome assemblies as described by van Hoek et al. [10].

### 2.5. Comparative Genetic Analysis

The assembled genome of the *stx_2f_*-carrying O157:H16 isolate (EH4279) was annotated with Prokka 1.14.6 and visualized using Proksee 1.1.1 [36]. The sequence was BLAST (BLAST+ 2.12.0 and BLAST Formatter 1.0.3)-compared with the genome of a publicly available *stx*-negative *E. coli* O157:H16 strain 98-3133 (Genbank accession number: CP051001) of ST10 [37] and the other *stx_2f_*-carrying O63:H6 strain, EH4183. The predicted prophages, including the *stx_2f_*-harboring prophage, were detected using Phigaro 2.3.0 [38]. The whole-genome Average Nucleotide Identity (ANI) was calculated with FastANI 1.3.3 [39]. The two *stx_2f_*-harboring prophages were characterized and compared to those of 167 human *stx_2f_*-carrying *E. coli* isolates based on the core Stx2f phage (cpMLST) scheme as described by van Hoek et al. [10].

## 3. Results

### 3.1. Stool Samples and Bacterial Isolates

The fecal sample collected on 26 October 2023 from the patient in this study tested positive for multiple gastrointestinal targets including *E. coli* O157 and *stx*_1/2_ at the laboratory of the AZ Turnhout (Table 1). This presumptive STEC-positive result was confirmed by the NRC STEC through isolation of a STEC strain (EH4183), which was characterized as non-sorbitol-fermenting non-O157, *stx_2f_*-positive, and *eae*-positive. STEC strain EH4183 showed selective growth on CT-SMAC and was susceptible to the tested antimicrobials.

The patients’ stool was still positive for both *E. coli* O157 and *stx*_1/2_ targets two weeks later (Table 1). The fecal sample was not referred to the NRC STEC for confirmation.

Yet, fifteen weeks later, an additional stool sample with a positive result for *E. coli* O157 and *stx_1/2_* targets was provided to the NRC STEC. The isolated STEC strain (EH4279) was reported as sorbitol-fermenting O157, *stx_2f_*-positive, and *eae*-positive (Table 1). No growth was observed on CT-SMAC. The isolate was susceptible to the tested antimicrobials.

### 3.2. In Silico Multilocus Sequence Typing Analysis and Identification of Genes Linked to Serotype, Virulence, and Antibiotic Resistance

The traditional typing results obtained for both *stx_2f_*-carrying strains, EH4183 and EH4279, were confirmed by whole-genome sequencing. The former isolate was characterized as O63:H6, ST583, with a repertoire of virulence genes including *stx_2f_* and several LEE genes (*eae* [α2-subtype], *espA*, *espC*, *espF*, and *tir*), non-LEE encoded effector *nleC*, the *cdtABC* genes–coding for the cytolethal distending toxin CDT type I—as well as the tEPEC determinant *bfpA* (Table 2 and Appendix A). The latter isolate was characterized as *stx_2f_*-positive O157:H16, ST10, with several LEE genes (*eae* [ε-subtype], *espA*, *espB*, *espF*, and *tir*), non-LEE encoded effectors (*nleB*, *nleC*), and the *cdtABC* genes (Table 2 and Appendix A). This isolate was negative for *bfpA*. The full repertoire of virulence genes identified in the genomes of both *stx_2f_*-carrying strains is presented in Appendix A.

Based on ResFinder analysis, no acquired antimicrobial resistance genes were found in both isolate genomes.

### 3.3. Comparative Genomics

The genome sequence of STEC O63:H6 strain EH4183 was assembled into three circular contigs: the 4,954,036 bp chromosome; a 163,213 bp IncFIB-type plasmid named pEH4183_1; and a 2753 bp plasmid named pEH4183_2 (Appendix A). Consistent with the PCR result indicating that the strain was *stx_2f_*-positive, the chromosome of strain EH4183 contained the *stx_2f_*-harboring prophage sequence–42.8 Kb in length (Appendix A). The plasmid pEH4183_1 contained the *bfpA* gene.

The genome sequence of STEC O157:H16 strain EH4279 was assembled into three circular contigs: the 4,834,419 bp chromosome; an 83,358 bp IncFII_pHN7A8_-type plasmid named pEH4279_1; and a 34,261 bp IncFII_29_-type plasmid named pEH4279_2 (Appendix A). Here also, the chromosome of strain EH4279 contained the *stx_2f_*-harboring prophage, predicted 44.3 Kb in length (Figure 1A).

The predicted *stx_2f_*-harboring prophage—a long-tailed phage belonging to the *Siphoviridae* family—was inserted adjacent to the *ssrA* gene on both the EH4183 and EH4279 genomes (Figure 1B). Moreover, the *cdtABC* gene cluster was located on the extremity of the *stx_2f_*-harboring prophage within EH4183 and EH4279 (Figure 1B).

A BLAST comparison showed that the STEC O157:H16 strain EH4279 shared a high nucleotide identity (99.97% ANI) to the diarrheagenic aEPEC O157:H16 strain 98-3133, but the latter displayed a gap of around 42 Kb largely corresponding to the *stx_2f_*-harboring prophage sequence. Contrarily, but not surprisingly, the chromosome of STEC O157:H16 strain EH4279 shared a lower overall nucleotide identity to the STEC O63:H6 strain EH4183 (96.69% ANI). Yet, the *stx_2f_*-harboring prophage sequence within EH4279 shared 99.6% identity over 96.1% coverage of the *stx_2f_*-harboring prophage sequence within EH4183. The upstream extremity of the prophage sequences with two coding DNA sequences (CDSs) predicted to encode for an integrase (*intA*) and a putative integrase (*gene1107* [Phigaro identifier]) showed no similarity (Figure 1B). Yet, adjacent to this region, both *stx_2f_*-harboring prophages shared identical integrase-coding sequence *int*Q.

It is remarkable that the two CDSs, *intA* and *gene1107*, located on the upstream extremity of the predicted *stx_2f_*-harboring prophage sequence within EH4279 were identical to two CDSs (*intA* and *gene4248* [Phigaro identifier]) in the genome of aEPEC O157:H16 strain 98-3133 (Figure 1B). Both CDSs were located on the extremity of a predicted *stx*-negative prophage integrated into the *ssrA* gene within 98-3133 (Appendix A). As no similar CDSs were found in the genome of EH4183 (Appendix A), they likely correspond to leftovers of an ancient *stx*-negative prophage similar to the predicted one found in aEPEC O157:H16 strain 98-3133, rather than being part of the integrated *stx_2f_*-harboring prophage (Figure 1B).

The high similarity of the *stx*_2f_-harboring prophages within EH4183 and EH4279 was confirmed by a comparative cpMLST analysis of 169 *stx*_2f_-harboring prophages from human STEC strains, which revealed that the *stx*_2f_-harboring prophages from EH4183 and EH4279 clustered together in a phylogenetic tree (Figure 2). More precisely, the analysis showed that 60 out of the 67 *stx_2f_*-prophage associated genes, according to van Hoek et al. [10], had identical alleles in both analyzed genomes of EH4183 and EH4279 (Appendix A). Yet, three genes were not found in the genome of EH4279. These three CDSs, present in EH4183 and located upstream of the integrase-coding sequence *intQ*, were predicted to encode for a transposase, an integrase, and a hypothetical protein. This observation matches the above-mentioned results.

Moreover, four genes differed, which encode for a phage tail tape measure protein (99.6% identity and 100% coverage), a minor tail protein—the phage minor tail protein L—(99.7% identity and 100% coverage), a phage tail fiber protein—the host specificity protein J—(96.6% identity and 100% coverage), and an attachment invasion locus protein precursor (99.5% identity and 100% coverage). Interestingly, the sequence differences at the position of the gene for the host specificity protein J were in the last distal portion of the gene.

## 4. Discussion

To the best of our knowledge, this is the first report of a *stx_2f_*-carrying *E. coli* O157:H16 isolated from a child with uncomplicated diarrhea, underlining, once more, the great capacity of *E. coli* to evolve due to gain of genes [13]. But identification of this Stx subtype in strains belonging to serotypes previously unknown to harbor *stx_2f_* has been reported before [40,41]. Subtype Stx2f is generally associated with a limited number of serotypes of, which serotype O63:H6 predominates in Belgium [8]. Until now, as far as we know, only one animal STEC O157:H16 strain, possessing the *stx_1_* and *stx_2_* genes, has been reported [32]. The latter strain should not carry the *stx_2f_* gene as the primers used by Ennis et al. [32] allowed amplification of all Stx subtypes, except Stx2f [42].

Sequencing data showed that the genome of the STEC O157:H16 EH4279 ST10 strain was closely related to that of an aEPEC O157:H16 strain (98-3133) of ST10 (LEE-positive, *bfp*A-negative, *stx*-negative). aEPEC O157:H16 are generally associated with mild disease. Sporadic and familial cases of diarrhea and bloody diarrhea have been reported [22,23,27,30]. However, one aEPEC O157:H16 has already been isolated from a HUS case, but it was unclear whether the HUS-associated *E. coli* O157:H16 strain lost *stx* prior to laboratory diagnostics or whether it was a coinfection with a *stx*-harboring strain that was the cause of the HUS, while STEC isolation was unsuccessful from the patient’s stool sample [26].

Since another *stx_2f_*-carrying *E. coli* strain, identified as STEC/tEPEC hybrid pathotype, with serotype O63:H6 was isolated from the same patient in this study fifteen weeks earlier, we speculated that the *stx_2f_*-encoding phage was transferred from this strain to the O157:H16 strain in the intestine of the present case. This hypothesis is plausible as *stx*_2_-harboring phages, released from STEC strains, infecting other *E. coli* strains have been described [13,19]. Based on comparative genomics, three lines of evidence support this hypothesis. First, the same integration site, near the *ssrA* gene, was identified for the predicted *stx_2f_*-harboring prophages obtained from both strains, EH4183 and EH4279. The *ssrA* locus was also present in the aEPEC O157:H16 strain 98-3133 examined in this study and represents a hot spot for the integration of phages into the bacterial genome of *Enterobacteriaceae* [15]. Second, the results in this study showed that the predicted *stx_2f_*-harboring prophage sequences obtained from both strains, EH4183 and EH4279, were highly similar (99.6% identity over 96.1% coverage). The uncovered region, located in the upstream extremity of the predicted *stx*_2f_-harboring prophage sequence within EH4279, contains an integrase-encoding gene (*intA*) and a gene annotated with a predicted function of integrase (*gene1107*). These CDSs shared 100% identity to two CDSs present in the aEPEC strain 98-3133, suggesting that these CDSs are part of the bacterial host genome, flanking the integrated *stx*_2f_-harboring prophage. This is possible as, adjacent to these genes, both *stx_2f_*-harboring prophages in EH4183 and EH4279 shared the identical integrase-coding sequence *int*Q, which is absent in strain 98-3133. Third, based on comparative cpMLST analysis, both prophage sequences clustered together and differed only by seven out of the 67 *stx_2f_*-phage associated genes. Indeed, three CDSs—coding for a transposase, an integrase, and a hypothetical protein—found on the extremity of the predicted *stx_2f_*-harboring prophage in EH4183 showed no significant similarity to those in EH4279. The additional differing four genes were predicted to encode tail-related proteins (96.6–99.5% identity and 100% coverage). A previous study demonstrated that the tail fiber gene encoding protein J, associated with host recognition, was highly conserved among most of the short-tailed phages [43]. Another phylogenetic study showed that both short- and long-tailed phages contained tail fiber genes sharing high nucleotide identities [44]. Yet, it has also been shown that mutations associated with a host range are located in the C-terminal part of protein J [45]. In the present study, we observed that the differences between both analyzed *stx*_2_-harboring prophages were also in the last distal portion of the gene encoding for protein J. We could emphasize that, under selective pressure, e.g., a prolonged period of intermittent diarrhea, these mutations enabled adaptation of the phage to a new bacterial host in the gut of the present case [46]. Taken together, these data largely support the fact that the *stx*_2f_-encoding phage was incorporated within the *E. coli* strain EH4279 in the intestine of the presented case during infection.

The other well-known virulence factor identified in the genome of EH4279, while absent in publicly available aEPEC O157:H16 strain 98-3133, is CDT type I. CDT type I has previously been reported in numerous human *stx*_2*f*_-carrying *E. coli* isolates [10]. In the EH4279 strain, the *cdtABC* gene cluster was located on the extremity of the *stx*_2*f*_-harboring prophage as reported previously for phages carrying genes for CDT type I [47]. To our knowledge, the association of CDT and *stx*_2f_ on a prophage has only been reported once for *stx*_2*f*_-carrying *E. albertii* isolated from humans and wild birds in the USA [15]. The production of CDT is thought to be associated with increased invasion, persistence, and disease severity in various bacterial pathogens [15]. So, in the present study, CDT type I could have contributed to the pathogenicity of EH4183 and EH4279.

It is a well-known fact that infections caused by *stx_2f_*-carrying *E. coli* strains tend to be underreported since certain molecular gastrointestinal assays for laboratory diagnosis of STEC infection do not detect *stx_2f_* [4,7]. Indeed, subtype Stx2f is the most divergent of the known Stx2 subtypes, showing very low homology with other subtypes [48]. Therefore, as stated already in previous reports, the inclusion of *stx*_2*f*_ primers can be of importance for the accurate diagnosis of STEC infections, certainly in patients with HUS.

Although the benefits of gastrointestinal assays have been demonstrated [49], the clinical interpretation of co-infections can be challenging. Indeed, in the present report, the gastrointestinal targets *E. coli* O157-*stx*_1/2_ were detected in combination with *Campylobacter* spp. and Sapovirus. A similar combination without *Campylobacter* spp. was obtained 2 weeks later, which is in line with previous reports showing that follow-up tests within 4 weeks of a positive test redemonstrate the initial pathogen(s) due to residual genetic material or colonization [49]. Of particular attention is the *E. coli* O157-*stx*_1/2_ combination that, once detected, was notified to the regional health inspection authorities, suggesting the presence of a highly virulent STEC of serotype O157:H7. Additionally, the *E. coli* O157-*stx*_1/2_ combination was detected over a time interval of 15 weeks, assuming that the child continued to shed STEC during this period of time. Prolonged shedding of STEC has been reported in children aged <6 years [50], questioning the need for the isolation of long-term STEC carriers to prevent its spread. Yet, most endemic STEC strains have a low pathogenicity and would not substantiate social exclusion stipulations [51]. So, the true composition of the clinical sample could be of importance for proportionate case investigation, balancing the risk of pathogen transmission against imposing unnecessary restrictions on cases.

In summary, we describe the first report of two *stx_2f_*-carrying *E. coli* strains with serotypes O63:H6 and O157:H16 isolated from a child with uncomplicated diarrhea. Comparative genomic analysis revealed that the STEC O157:H16 strain was closely related to aEPEC O157:H16 strain 98-3133. The *stx_2f_*-harboring prophages of the two distinct STEC strains isolated from the present case were highly similar, suggesting that the *stx_2f_*-harboring phage might have been transferred from the STEC/tEPEC O63:H6 strain to the aEPEC O157:H16 strain in the gut of the child.

## Figures and Tables

**Figure 1 pathogens-13-01002-f001:**
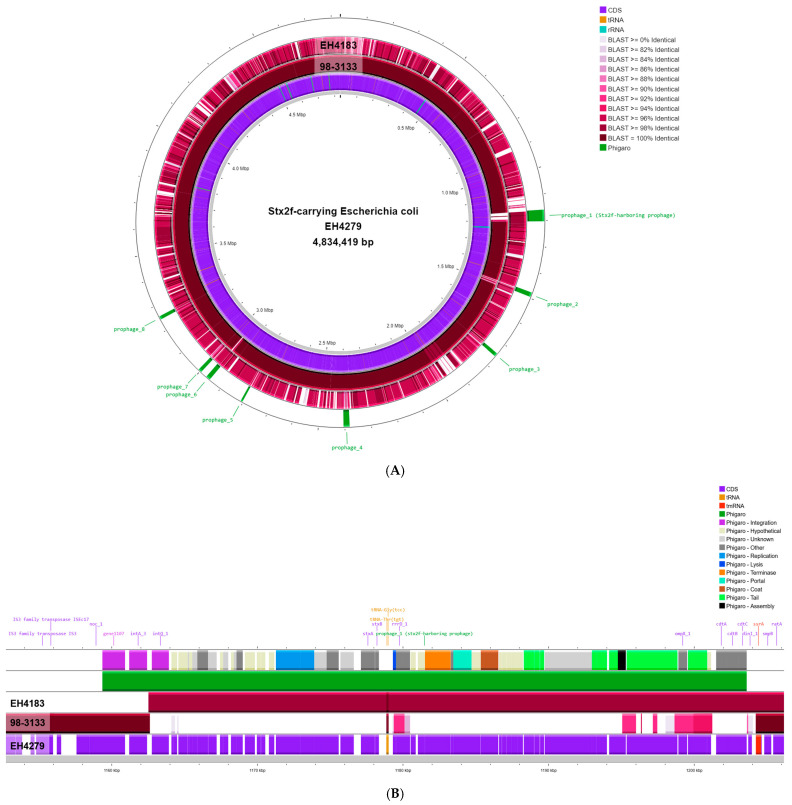
BLAST genome comparison of STEC O157:H16 strain EH4279 with aEPEC O157:H16 strain 98-3133 and STEC O63:H6 strain EH4183, with indication of the predicted prophages. (**A**). Chromosome comparison of 98-3133 to EH4279 using BLAST demonstrates a gap of around 42 Kb in 98-3133, largely corresponding to the *stx_2f_*-harboring prophage sequence. Conversely, this region is highly similar in both sequences of EH4183 and EH4279. The aligned genomes of 98-3133 and EH4183 are colored by sequence identity, ranging from 0 to 100%, using BLAST Formatter. (**B**)**.** Closer view on the *stx_2f_*-harboring prophage and its surrounding region. The predicted *stx_2f_*-harboring prophage detected in the EH4279 genome using Phigaro is shown in dark green. The prophage genes (top track) are color-coded based on their biological functions. Hypothetical proteins are not labeled in the present figure.

**Figure 2 pathogens-13-01002-f002:**
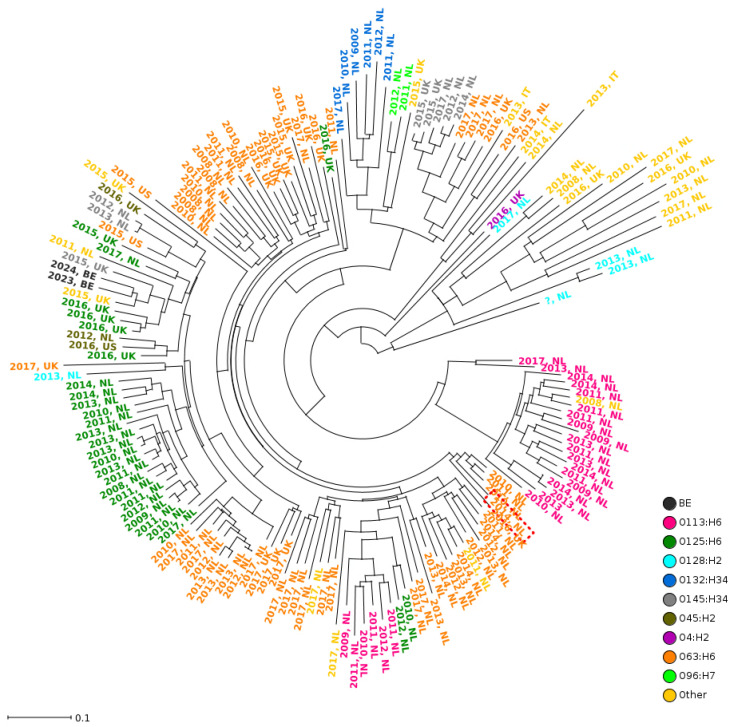
Neighbor-joining phylogenetic tree of 169 human *stx_2f_*-carrying *E. coli* isolates based on the core Stx2f phage MLST (cpMLST) data. The isolates are color-coded based on serotype except for the two black-colored isolates of this study, O63:H6 EH4183 (2023, BE) and O157:H16 EH4279 (2024, BE). In addition, the isolates are labeled by year and country of isolation.

**Table 1 pathogens-13-01002-t001:** Laboratory results obtained for the fecal specimens collected from the patient in this study.

Date of Sampling	Gastrointestinal Targets Detected in Stool ^1^ (Cycle Threshold [Ct]-Value)	Isolate Characteristics (Identifier) ^2^
26 October 2023	*E. coli* O157 (33.16), *stx*_1/2_ (28) *Campylobacter* (36 ^3^), Sapovirus (26)	Non-O157 *stx_2f_* + *eae* + (EH4183)
6 November 2023	*E. coli* O157 (28.47), *stx*_1/2_ (26) Sapovirus (27)	No isolate ^4^
12 February 2024	*E. coli* O157 (37.69), *stx*_1/2_ (36) Adenovirus (25 ^5^), Cryptosporidium (30)	O157 *stx_2f_* + *eae* + (EH4279)

^1^ Results obtained with the Allplex^TM^ GI-EB Screening Assay, the Allplex^TM^ GI-Virus Assay, and the Allplex^TM^ GI-parasite Assay (Seegene, Seoul, Republic of Korea); ^2^ results obtained with the conventional NRC STEC tests; ^3^
*Campylobacter* culture-negative; ^4^ sample not referred to the NRC STEC; ^5^ adenovirus culture-positive.

**Table 2 pathogens-13-01002-t002:** Characterization of both *stx_2f_*-carrying *E. coli* isolates, with indication of phylogroup, pathotype, serotype, the sequence type (ST), and major virulence genes.

Strain	Phylogroup	Pathotype	Serotype	ST	Major Virulence Genes
EH4183	B2	STEC/tEPEC	O63:H6	583	*stx_2f_*, *eae* (α2-subtype), *cdtABC*, *bfp*A
EH4279	A	STEC/aEPEC	O157:H16	10	*stx_2f_*, *eae* (ε-subtype), *cdtABC*

## Data Availability

The sequences of *stx_2f_*-carrying *E. coli* O63:H3 are available online under BioSample numbers: SAMEA115376172 (Illumina FASTQ files), SAMN43992296 (Oxford Nanopore Sequencing data), and SAMN44323898 (hybrid assembly). The sequences of *stx_2f_*-carrying *E. coli* O157:H16 are available online under BioSample numbers: SAMEA115903504 (Illumina data) and SAMN43992297 (Oxford Nanopore Sequencing data) and SAMN44323899 (hybrid assembly). All supporting data and protocols have been provided within the article or through Appendix A. Appendix A are available with the online version of this article.

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
