# Peer review of "Intestinal Carriage of Two Distinct stx2f-Carrying Escherichia coli Strains by a Child with Uncomplicated Diarrhea"

_pathogens, 2024, doi:10.3390/pathogens13111002_

Round 1

Reviewer 1 Report

Comments and Suggestions for Authors

Crombe et al. present a nice study showing the most probable transfer of the stx2f gene via phages  from one E.coli to another, different serotype in a child with recurring diarrhoea. The paper is well written and the data convincing. Of course, one can never know, whether there could have been another E.coli or other intermediate-bacterial host, transferring the phage/gene to the second E.coli, but since it occurred in the same patient, it seems to be a probable fact.

I have no other general comments to the study, but a few minor comments.

We hear that no antibiotic resistance genes were detected, but since an AST was performed it might be relevant to describe, that phenotypic tests also showed no resistance?

Line 121: misspelling of Ttyping

Lines 158 ff: Two drugs are mentioned, Enterol and Tiorfix. Since i and probably other non-Belgians do not know what they are, the authors should reveal what these medicines contain/represent?

Author Response

Comments 1: We hear that no antibiotic resistance genes were detected, but since an AST was performed it might be relevant to describe, that phenotypic tests also showed no resistance?

Response 1: We rephrased this information on line 169 and 176.

Most STEC of serotype O157:H7/H- are resistant to tellurite, while the very low concentrations of cefixime present in CT-SMAC medium inhibit other non-sorbitol fermenting Enterobacteriaceae but not E. coli (without resistance determinants) (see Chapman et al 1991; doi:10.1099/00222615-35-2-107). CT-SMAC medium is often used to facilitate isolation of O157:H7/H- and some other serotypes of STEC. The concentrations of tellurite and cefixime were added (lines 104-105).

The AST was performed according to EUCAST: see M&M, lines 109-114 and results, lines 169-170 and line 176.

Comments 2: Misspelling of Ttyping.

Response 2: We updated this information on line 140 (previously line 121).

Comments 3: Two drugs are mentioned, Enterol and Tiorfix. Since i and probably other non-Belgians do not know what they are, the authors should reveal what these medicines contain/represent?

Response 3: We updated this information on line 90 and 93 (previously line 158).

“Fifteen weeks later, the child consulted the pediatrician with complaints of intermittent vomiting, diarrhea for 4 days and stomach cramps. The child was treated with probiotic Enterol® (Saccharomyces boulardii) and oral rehydration solution. The child returned to the pediatrician five days later with complaints of watery diarrhea. The child was treated symptomatically with antidiarrheal Tiorfix® (10 mg racecadotril/sachet) and recovered promptly. A fecal sample was collected on 12 February 2024.”

Reviewer 2 Report

Comments and Suggestions for Authors

This manuscript examines two Escherichia coli (E. coli) strains carrying the stx2f gene, isolated 15 weeks apart from a child with diarrhea. The researchers used short- and long-read sequencing to analyze the genetic relationship between the two strains, revealing similarities and potential gene transfer. The study provides some data to enhance the understanding of the genetic dynamics and virulence evolution of E. coli, emphasizing the possible key role of phages in bacterial adaptation.

It is recommended to explore environmental or dietary factors influencing gene transfer, and incorporating the child’s specific dietary information would add more value. Additionally, two suggestions are provided:

  1. Section 3.1 (Case description) should be moved to the Materials and Methods section.
  2. In the references, the Latin names of bacteria should be italicized.

Author Response

Comments 1: It is recommended to explore environmental or dietary factors influencing gene transfer, and incorporating the child’s specific dietary information would add more value.

Response 1: No dietary habits were reported on the NRC STEC. As no diet was specified, we could not update this information accordingly.

Comments 2: Section 3.1 (Case description) should be moved to the Materials and Methods section.

Response 2: We adapted the text accordingly.

Section 2.1. Case description (line 82)

Section 3.1. Stool samples and bacterial isolates (line 164)

Comments 3: In the references, the Latin names of bacteria should be italicized.

Response 3: We updated this information in the References section.

Reviewer 3 Report

Comments and Suggestions for Authors

In this manuscript, Crombé et al. collected samples from child with uncomplicated diarrhea for the isolation and genomic characterization of stx2f-carrying Escherichia coli, followed by the genomic analysis and resistome profiling. The subject in this case report matches well the scope of this journal, and the results are supported by the performed assay. This reviewer finds significance and novelty in this work.  However, there are few errors that need to be addressed before publication. There are many literal errors and disagreements in the text. Thus, this reviewer suggests that the authors carefully check for basic errors and discrepancies.

Line 19: The word uncomplicated diarrhea is difficult to understand; please elaborate it in the abstract and methodology section Antimicrobial resistance is a major issue regarding public health. Please mention AMR relevant data in the abstract part

Line 49-52: This is hard to understand these lines. Please rephrase this sentence for more understanding. Please add a few more details about the global emergence of STEC and AMR patterns, which makes it more difficult to treat along with its pathogenesis. Authors can get information from the following articles (https://doi.org/10.1016/j.csbj.2023.12.041,https://doi.org/10.1128/spectrum.02113-22, https://doi.org/10.3389/fmicb.2022.1018682.)

Line 82-95: This chapter (2.1. bacterial isolates) should be rewritten. Please elaborate case history in this section instead of results section.

Line 97: This chapter 2.2, you can add DNA extraction and whole genome sequencing

Line 184. Please rephrase this line and discuss these results in the discussion section. What will be the possible reason for the absence of AMR genes? Are these authors testing the phenotypic resistance of these isolates? Please evaluate association of genotypic and phenotypic antimicrobial resistance.

Figure 1: Quality of Figure is very low. Its very difficult to read

Figure 2: Illustration of figure can be improved.

Comments on the Quality of English Language

There are some minor issues

Author Response

Comments 1: Line 19: The word uncomplicated diarrhea is difficult to understand; please elaborate it in the abstract and methodology section.

Response 1: Uncomplicated diarrhea is generally used for STEC cases of non-bloody diarrhea (doi : org/10.1086/338115; doi: 10.1128/JCM.40.4.1441-1446.2002). Stx2f is mainly present in strains isolated from patients with uncomplicated diarrhea (doi: 10.3390/microorganisms9112374). The information provided to the NRC STEC has been listed in section 2.1 (lines 82-94).

Comments 2: Antimicrobial resistance is a major issue regarding public health. Please mention AMR relevant data in the abstract part.

Response 2: We are also concerned by antimicrobial resistance. However, as this is not the main topic of this paper and as the use of antibiotics in individuals with STEC infections is not recommended (doi: 10.1093/cid/ciw099), we did not add information about AMR in the abstract. Nevertheless, more information was added about AMR, see Response 4.

Comments 3: Line 49-52: This is hard to understand these lines. Please rephrase this sentence for more understanding.

Response 3: The sentence was rephrased. In the Annual epidemiological report for 2021 of ECDC no numbers are specified, only the percentage of 47,0% is provided.

« During this period, the dominant stx2f-positive E. coli serotype was O63:H6 (67.1%; 51/76). The same was observed, in 2021, in the European Union/European Economic Area (EU/EEA) where 47.0% of the stx2f-carrying E. coli isolates were serotyped as O63:H6 (data from 23 EU/EEA countries) [9] ».

Comments 4: Please add a few more details about the global emergence of STEC and AMR patterns, which makes it more difficult to treat along with its pathogenesis. Authors can get information from the following articles (https://doi.org/10.1016/j.csbj.2023.12.041,https://doi.org/10.1128/spectrum.02113-22, https://doi.org/10.3389/fmicb.2022.1018682.).

Response 4: More information was provided about antimicrobial resistance: phenotypic AST was performed according to EUCAST: see M&M, lines 109-114 and results, lines 169-170 and line 176.

Further, the use of ResFinder 4.5.0 is already mentioned at line 146. Lines 197-198 mention that no acquired antimicrobial resistance genes were detected.

We agree that antimicrobial resistance is a major public health threat, in particular in zoonotic pathogens where the occurrence of multidrug resistant strains is increasing. Resistant strains of gastrointestinal pathogens may have a selective advantage over other bacteria in intestines of humans or animals undergoing antibiotic treatments. However, as the use of antibiotics in individuals with STEC infections is not recommended (doi: 10.1093/cid/ciw099), we think that adding more comments about AMR is not needed in this paper dealing with virulence factors transferred between two E. coli strains. The interested reader will observe that no AMR is involved here.

Comments 5: Line 82-95: This chapter (2.1. bacterial isolates) should be rewritten. Please elaborate case history in this section instead of results section.

Response 5: We adapted the text accordingly.

Section 2.1. Case description (line 82)

Section 3.1. Stool samples and bacterial isolates (line 164)

Comments 6: Line 97: This chapter 2.2, you can add DNA extraction and whole genome sequencing.

Response 6: We adapted the text accordingly (line 116).

Comments 7: Line 184. Please rephrase this line and discuss these results in the discussion section. What will be the possible reason for the absence of AMR genes? Are these authors testing the phenotypic resistance of these isolates? Please evaluate association of genotypic and phenotypic antimicrobial resistance.

Response 7: The sentence was rephrased (see lines 197-198). Multi-susceptible STEC are not exceptional (https://doi.org/10.1016/S0168-1605(00)00351-2). See Response 4 for further comments.

Comments 8: Quality of Figure is very low. Its very difficult to read.

Response 8: The figures cannot be enlarged however by zooming on the figure it is possible to have a detailed look at the figure.

Comments 9: Illustration of figure can be improved.

Response 9: The illustrations of figures cannot be enlarged however by zooming on the figure it is possible to have a detailed look at the figure.